# Nanoparticle Assisted EOR during Sand-Pack Flooding: Electrical Tomography to Assess Flow Dynamics and Oil Recovery [note 1]

**DOI:** 10.3390/s19143036

**Published:** 2019-07-10

**Authors:** Phillip Nwufoh, Zhongliang Hu, Dongsheng Wen, Mi Wang

**Affiliations:** School of Chemical and Process Engineering, University of Leeds, Leeds LS2 9JT, UK

**Keywords:** imaging techniques, multiphase flow, nanoparticles, enhanced oil recovery, tomography, sand-pack flooding

## Abstract

Silica nanoparticles have been shown to exhibit many characteristics that allow for additional oil to be recovered during sand-pack flooding experiments. Additionally various imaging techniques have been employed in the past to visually compare flooding procedures including x-ray computed tomography and magnetic resonance imaging; however, these techniques require the sample to be destroyed or sliced after the flooding experiment finishes. Electrical resistance tomography (ERT) overcomes these limitations by offering a non-destructive visualization method allowing for online images to be taken during the flooding process by the determination of spatial distribution of electrical resistivity, thus making it suitable for sand-packs. During the scope of this research a new sand-pack system and methodology was created which utilized ERT as a monitoring tool. Two concentrations, 0.5 wt% and 1.0 wt%, of SiO_2_ nanoparticles were compared with runs using only brine to compare the recovery efficiency and explore the ability of ERT to monitor the flooding process. Electrical resistance tomography was found to be an effective tool in monitoring local recovery efficiency revealing 1.0 wt% SiO_2_ to be more effective than 0.5 wt% and brine only runs during the scope of this research. A new method involving the slope function in excel was used to compare the effects of nanofluids on resistivity trends also revealing information about the rate of recovery against time. SiO_2_ nanofluid recovery mechanisms such interfacial tension reduction and viscosity enhancement were then considered to explain why the nanofluids resulted in greater oil recovery.

## 1. Introduction

Both electrical conductivity and resistivity measurements have been conducted in the past to characterize porous media flow, especially during soil infiltration measurements [1,2,3]. The resistivity of a porous media is governed by the electrical resistivity of the constituent phases, saturation content, fabric, and porosity. These resistivity measurements have been conducted to a lesser extent in regards to flooding processes for oil recovery. Local resistivity distributions can provide information on phase behavior and recovery efficiency during sand-pack flooding experiments for nanoparticle induced, enhanced oil recovery. Typically x-ray computed tomography and magnetic resonance imaging are used to obtain images of flow patterns during flooding experiments. Although these options provide better spatial resolutions, electrical resistance tomography (ERT) also has many benefits including excellent temporal resolutions, great practicability, and its ability to perform reconstructions in real time.

Flooding experiments for enhanced oil recovery typically consists of a porous media that’s largely saturated with oil, which is then flushed with brine until there is a plateau in oil recovery. Once all of the movable oil is recovered, a fraction of bypassed oil will still be present within the pores in what is termed residual oil. To reduce the residual oil saturation, different enhanced oil recovery techniques must be applied including thermal, gas, and chemical injections. The scope of this research focuses on a branch of chemical injection termed nano-EOR in which nanoparticles are introduced to increase oil recovery after the water-flooding stage is deemed to be no longer efficient. Nanoparticles have many desirable characteristics which allow for a reduction in residual oil saturation, including (i) their ability to travel through the smallest of pore throats due to their minute size (1–100 nm) which allows them to penetrate deep into reservoir rocks, (ii) their large specific surface area, resulting in an increased contact area of nanoparticles with the oil phase, and better interaction between phases [4], (iii) their high multi-functionality allowing them to be fabricated for a specific task, thus allowing for nanoparticle characteristics such as particle coatings and morphology to be altered [5,6,7], and lastly, (iv) their ability to change both multiphase and rock–fluid behaviors, allowing for the interfacial tension and wettability of the system to be altered [8,9,10,11].

There are many experimental challenges associated with using ERT as a monitoring tool during nano-EOR flooding experiments. Firstly, a sand-pack setup had to be constructed which incorporates ERT around the perimeter of the vessel. Then the limitations of ERT itself had to be considered such as the lack of spatial resolution (5% of vessel diameter) which meant the nanoparticles themselves could not be detected during the measurements as the resolution in this case is on the millimeter scale, and the nanoparticles on the nano-scale. Therefore, instead of considering the flow behavior of the nanoparticles themselves, the change in flow behavior or oil recovery rate resulting from the presence of nanoparticles will be considered and if large enough should be able to be detected by ERT. A major limitation of ERT is the need for the medium to be conductive so an electrical continuity exists between the sensors and the medium. Previously, researchers have considered conductive flows through porous media such as water through soil. However performing such experiments with oil is more challenging due to the electrically discontinuous nature of oil making acquiring a signal much more difficult. The advent of new sensing methods, particularly the conductive ring and resistor network arrangement, have increased the measurement range for ERT allowing for more accurate measurements in highly intermittent environments such as oil saturated porous medias.

The aims here are to construct a sand-pack setup which incorporates traditional ERT sensors around the sand-pack. Then ERT will be used to monitor oil recovery during nano-EOR flooding experiments, and to validate the oil recovery results using the local resistivity plots and tomograms. Additionally, ERT will also be used to reveal any differences in flow behavior and oil recovery brought about by the presence of nanoparticles by observing changes in local resistivity trends and conductivity profiles with time.

## 2. Materials and Methods

### 2.1. The System

A sand-pack flooding system was developed in order to utilize tomography as a visualization tool to determine the effect of nanofluid on porous flow (Figure 1). In order to achieve this a column made of perplex (5 cm diameter) was fitted with dual plane ERT sensors and filled with building sand to represent the porous media. A vibrating electric oscillator (50 Hz), capable of variable amplitude settings (max amplitude >0.15 mm) was situated underneath the column to provide a well compacted sand-pack. A gravity feed system was developed in order to introduce the fluids at a constant pressure head, and solenoid timer valves allowed for pulses of brine, with and without nanoparticles, to be introduced. The solenoid valves were opened for periods of 5 min at a time, after a period of 1 h and 15 min, during which portions of about 10 mL of fluids were released from the medical drainage bags. A collection table was situated beneath the table consisting of a number of beakers arranged on a collection tray, which was placed on top of a rotation motor. The rotation angle, time spent at each angle, and number of times that the cycle is completed, could all be set by the user making it suitable for different flooding scenarios.

The dual plane ERT sensors consisted of 16 electrodes (12.7 mm × 25.4 mm) made from stainless steel which were connected to the data acquisition system via 36 pin connectors (IEE-488) on the front of the system. The data acquisition system was the P2000 ERT system, a system designed by Industrial Tomography Systems, based in Manchester, UK. For the scope of these experiments, an injection current of 15 mA was used and a frame capture rate of 1 fps was adequate to capture the flooding process over the range of 4.5 h. An adaptor cable was used to create a more homogenous sensitivity distribution by mimicking the technology incorporated in resistor-network and conductive ring sensing apparatuses [11]. This allowed for the measurement of more complex multiphase flows with an electrical discontinuous phase such as the three phase oil-brine-nanofluid flows considered in this setup.

### 2.2. Gravity Feed System

In order for the system to work effectively, the flow rate must be near identical for both phases and kept constant as a large change in flowrate could alter the conductivity distribution within the sensing plane, thus making the results invalid. Conventionally, the use of an injection pump along with holding vessels for each fluid is used in flooding setups to introduce various fluids. However, these setups are relatively expensive and require the valves to be opened and closed manually when changing flooding fluids. The use of a gravity feed system along with timer valves allows the fluids to be introduced automatically at near constant flow rates, allowing for 5 min pulsations of each flooding fluid to be introduced without having to manually open and close the valves during every pulsation. The gravity feed was simply constructed by connecting 2000 mL medical drainage bags (Beambridge Medical Ltd., Guildford, UK) to 10 mm tubing which fed to the inlet chamber via timer valves. The pressure drop was calculated by using measuring tape to quantify the hydraulic head in feet, and converted to atmospheres using a conversion factor where 304.8 mm = 2989.1 Pa. The pressure drop for 3091 mm was found to be 38,260 Pa, which was used for all the experimental runs.

A liquid or object that is free to move usually moves spontaneously from a state of higher potential energy to a state of lower potential energy. The same applies for water in a porous medium, such that a unit volume or mass of water will tend to migrate from an area of higher potential energy, such as in Figure 2, highlighting the total heads at the inlet and outlet. The driving force equation is used to explain the water flow from the drainage bags to the outlet, and quantify the force of displacement, given by the total potential (HH) at two points (HH_A_, HH_B_) divided by the distance (L_AB_) between the points:(1)$$df=-\frac{dHH}{dx}=\frac{dHH_{A}-dHH_{B}}{L_{AB}\;}$$

Amending the above equation and solving for driving force gives:(2)$$df=-\frac{dHH}{dx}=\frac{dTH_{TOP}-dTH_{BOTTOM}}{L_{AB}\;}\;\mathrm{df}=\frac{3231-450}{400}=6.95$$

### 2.3. Vibrating Oscillator

The packing method is a crucial factor for EOR-flooding affecting the nature of the flow, porosity, and permeability. Almost all reservoir rocks are composed of sedimentary rocks in the porosity range of 10–40% in sandstones and 5–25% in carbonates, whilst permeability is found to vary much more from a fraction of a millidarcy to several darcies. In order to provide a packing that is representative of reservoir values, an eclectic oscillator was incorporated into the setup so that the compaction could take place without the user having to hold the column over a vibrating mechanism, which is how a sand-pack is conventionally compacted. Also when the user is manually holding a sand-pack, there is a higher possibility of error in packing as the user will change the position of the pack, making them less uniform and prone to heterogeneities in different layers. The vibrating oscillator being incorporated into the setup overcomes these limitations as the column is always fixed in the same position during compaction and the compaction time can last for much longer as the user does not have to hold the column. Therefore, compaction can last several hours and the user can return when the packing is complete, which also ensures the packing is repeatable and uniform before each run. The vibrating oscillator itself (model JT-51B) ran on a frequency of 50 Hz and was capable of variable amplitude settings (max amplitude ≥15 mm). The maximum amplitude was used during the range of the experiments and the compaction time was kept to 1.5 hours. The repeatability of the packing method is shown in Figure 3 below which reveals the relationship between permeability and compaction time. The packing method was found to be highly repeatable with a maximum error of 6% over three runs at 15 min compaction time, which lowers as the compaction time increases to eventually less than 2% error after 90 min of compaction.

### 2.4. Effluent Collection System

One of the main challenges faced when constructing this particular sand-pack column was how to quantify the recovered oil. Traditional effluent collection methods include graduated pipettes, fraction collectors, mass flow meters, and digital scales [12]. A new effluent collection system and methodology was developed based on the concept of a fraction collector.

A rotating turntable, purchased from Comxim Ltd., was the base of the effluent collection table and acted as a rotating motor. A collection tray was then fitted on top of the rotating motor which housed a number of collection beakers. The beakers were arranged in such a way so that no fluid was lost when they changed position (Figure 4). The table’s rotational settings could be altered so that the rotation angle, time spent at each angle, and number of times the cycle is completed could all be set before the experiment. The exact quantity of oil was measured by pouring fluids from the collection beakers into graduated cylinders after the experiment. A small amount of excess oil is left on the sides of the beakers which is also accounted by simply subtracting the mass of an empty beaker from the mass of the beaker with excess oil, and converting to volume since the density of oil is known. The details of the rotating tunable are listed in Table 1.

### 2.5. Sand-Pack Flooding Preparations

Electrical Resistance Tomography (ERT) works by obtaining a conductivity/resistivity distribution in the domain of interest by using electrodes that induce currents or voltages, and electrodes that measure the resultant currents or voltages. Therefore, in order to effectively visualize the effect of nanofluids on oil recovery, the conductivities of both the brine phase and nanofluid phase must be near identical so any changes in conductivity distribution are a function of oil recovery and not the conductivity contrast between the two flooding fluids. To achieve this, small quantities of NaCl solution were added to each phase to ensure their conductivities matched. The tracer dilution method was used to quantify how much NaCl solution is needed to be added to the tap water and nanofluid phases, given by:(3)$$\Delta t\;\left(C_{0}\;q_{v}+C_{1}\;q_{1}\right)=\Delta t\;\left(C_{2}(q_{v}+\;q_{1}\right))$$
where Δt is the change in time, $C_{0}$ is the initial conductivity of DI water, $C_{1}$ is the conductivity of the NaCl solution to be added to the DI water, $C_{2}$ is the post mixing conductivity of the brine, $q_{v}$ is the volumetric flow before mixing, and $q_{1}$ is the volumetric flow after mixing.

Since, $q_{v}=\;\frac{volume\;}{time}$, the equation above can be rewritten in terms of volume: (4)$$C_{0}V_{v}+C_{1}V_{1}=C_{2}\;\left(V_{v}+V_{1}\right)$$
where $V_{v}$ is the total volume of DI water before mixing and $V_{1}$ is the volume of NaCl solution to be added to DI water.

The conductivities of both the brine and nanofluid phases were matched using Equation (4), and a Eutech CyberSca PC 6500 bench conductivity meter was used to validate the results. The findings are revealed in Table 2.

Next, the sand was prepared by cleaning and drying it before saturating it with known volumes of oil and water. This was achieved with prior knowledge of the average pore volume for a range of sand-packs as a function of compaction time and intensity. The porosities of the sand-packs were found to be in the range of 35–37% and the permeability between 5–6 Darcy’s (Table 3). Once the average pore volume was obtained (found to be approximately 282.42 mL), the sand was mixed with 1/3 water and 2/3 oil to fill this pore volume and ensure the sand was fully saturated. If the sand is not fully saturated, the presence of air bubbles could show as areas of low conductivity on the tomograms and could be mistaken for oil. The compaction process was then initiated by switching on the electric oscillator and leaving it for 1.5 h. Lastly, before the fluids were introduced, the parameters were set for the ITS P2000 system, solenoid timer valves, and the rotating collection table.

### 2.6. Sand-Pack Flooding Procedure

The flooding procedure differs slightly from the conventional method as the sand-pack is already saturated with known volumes of oil and water prior to packing. Since the initial water and oil saturations were already known, the traditional drainage process was skipped, where oil is injected into the sand-pack, and instead flooding began at the first forced imbibition process in which brine was injected at a constant rate of 2 mL/min for 0.8 pore volumes (PVs) to displace the oil phase. Next the nanofluid pulsations were introduced at the same flow rate by setting the timer of valves to allow for 5 min intervals of nanofluid and brine for 1.3 PVs. The accumulated oil was then collected in a number of beakers housed on a collection tray which rotated automatically in timed intervals. This allowed for the cumulative oil recovery to be accounted automatically without having to observe the recovery in timed increments over the duration of 4.5 h. This coupled with the gravity feed system to introduce the fluids meant that the whole flooding process could be run automatically, with the fluids being introduced as well as the collected in timed increments without the user having to be present. When the flooding was complete, the recovered oil was then accounted for in each beaker by pouring their contents into a graduated cylinder as described in Section 2.4.

### 2.7. Materials

De-ionized water was used as the base fluid for dispersing the nanoparticles, as well as the formation liquid for all the experiments. Laboratory grade sodium chloride purchased from Sigma Aldrich was used to make the brine solutions. The density of the brine was measured at 1000 ± 0.01 g/cm^3^, pH 6.46 ± 0.3, and the dynamic viscosity 0.87 ± 0.01 mPa·s at 25 °C. The oil phase was a light mineral oil from Fisher Scientific with a density of 0.84 g/cm^3^ and a dynamic viscosity of 39.95 mPa ± 0.11 mPa·s at 25 °C. Silica nanoparticles with a diameter of 80 nm were purchased from get nanomaterials and were used for all experiments. The sand used was a fine building sand purchased from Wickes with an average grain size between 2–4 mm.

## 3. Results and Discussion

### 3.1. Flow Imaging

Relative changes of conductivity as a function of time were used to visualize the flow behavior of oil, water, and nanofluid phases as flooding progresses to provide information on local recovery efficiency. The tomograms shown in Figure 5 began with a relatively high concentration of oil phase (in blue color), which was shown to be displaced by the aqueous phase (in red color) over time. During both runs the concentration of oil was initially seen to deplete from the center of the column, leaving behind zones of bypassed oil towards the outer perimeter as time progressed. The conductivity scale was selected between 0.98 mS/cm and 1.14 mS/cm to ensure there was an adequate contrast to differentiate between the low conductivity phase (oil) and the high conductivity phases (brine and nanofluids). To the left of the tomograms are 15 min time stamps highlighting relative changes in conductivity as time progressed. 

The objective of the study was to ultize the ERT to qualitatively compare the recovery efficiency during different flooding scenarios. Figure 5 above reveals the tomograms for brine-only, 0.5 wt% SiO_2_, and 1.0 wt% SiO_2_. All three of the tomogram sets show zones of bypassed oil after the initial water flood, leaving a good opportunity to observe the effects of silica nanofluid injection. The run using only brine clearly shows a much lower recovery efficiency along the measuring planes, when compared with both runs using silica nanofluid. For the brine only run, after 1 h and 15 min, the local oil recovery becomes almost stagnant in both planes and does not show large variations in flow behavior and flooding efficiency. When this run is compared with the silica nanofluid runs, a different picture is painted revealing the ability of nanofluids to open new flow pathways and alter the rate of recovery. The nanofluids were introduced after 1 h and 15 min during both nanofluid runs. When considering the 1.0 wt% run, it is apparent that the nanofluid injection displaced some residual oil below the sensors which then accumulated in oil pockets at the 4 o’clock position. The tomograms for 0.5 wt% SiO_2_ show two zones of residual oil at the 1 o’clock and 11 o’clock positions, which are both gradually stripped away, until only a very faint pocket of oil can be seen at the 1 o’clock position. The run using 1 wt% SiO_2_ reveals the nanofluids to be more effective in removing local oil residue than both the brine only and 0.5 wt% nanofluids, as confirmed by the resistivity plots (Section 3.2).

The tomograms are validated by the resistivity and slope function plots. There is a sudden change in recovery efficiency between 3 h 15 min and 3 h 30 min, in which the flow channels begin to open and spread laterally. This rapid change is believed to be the result of nanoparticles altering the flow pattern and allowing flooding fluids to infiltrate pore channels which were previously bypassed. However, other recovery mechanisms could explain the differences in flow structure (Section 3.5). These quick changes in the tomograms are also related to the plots for resistivity and cumulative oil recovery, with the resistivity plot showing quick variations at the same time and the plot cumulative oil recovery plot showing additional recovered oil between 1.2–1.5 pore volumes (PVs) (between 180–210 min) for the 1.0 wt% SiO_2_ and between 1.2–1.6 PVs (between 200–240 min) with regards to the 0.5 wt% SiO_2_. This behavior can be attributed to a number of recovery mechanisms such as interfacial tension (IFT) reduction, viscosity enhancement, or the presence of nanoparticles altering the pore network and flow structure of the sand-pack. A recovery analysis was conducted later in the paper to determine the influence of some of these mechanisms. The conductivity profiles are also displayed in Figure 6, Figure 7 and Figure 8 and reveal the changes to flow dynamics with time across the central region of the tomograms. The effects of nanofluids on oil recovery are highlighted by revealing larger increases in conductivity trends after the introduction of nanofluids which were not present during the runs without nanofluids. The trends show conductivity to increase as the flooding progresses and oil is displaced, followed by a near convergence of the trends as recovery efficiency begins to stagnate. The plots reveal the ability of ERT to add to the physical understanding of the flow dynamics and oil displacement process with time.

### 3.2. Resistivity vs. Time

The ability of ERT to monitor the local flooding efficiency of different fluids was assessed during the scope of the experiments by considering resistivity plots as a function of time (Figure 9). The resistivities were found to gradually decrease during all the runs as oil was displaced from the sensing planes and replaced with brine. From the data it is evident that the local recovery efficiency was highest during the 1 wt% SiO_2_ run, followed by the 0.5 wt% SiO_2_ run and, lastly, the brine only run. The effects of nanofluids are clearly shown by the gradients of the plots, with both the 0.5 wt% and 1.0 wt% runs revealing sudden downward trends after the introduction of nanofluids which were not present during the brine runs. The plot also reveals 1 wt% SiO_2_ to have a quicker and slightly more profound effect on oil recovery than the 0.5 wt% SiO_2_ run. For example, taking run no.1 into account, the sudden downward trend during the 1 wt% SiO_2_ run occurs approximately 55 min after the introduction of nanofluids, whilst the downward gradient occurs approximately 115 min after nanofluid injection. Additionally, since one pulsation of nanofluid is approximately 10 mL, the volume of flooding fluids before a change in gradient could be approximated, which was found to be 5.5 pulsation cycles of brine/nanofluids each (where every pulsation = 10 mL) for the case of 1 wt% SiO_2_ and 12 pulsation cycles of brine/nanofluid for the 0.5 wt% SiO_2_ run. The quicker rate of recovery may be a combination of IFT reduction, nanofluid viscosity enhancement, or temporary log-jamming. Another important feature to note is the local resistivity curves correlate well with the oil recovery data. For the brine only runs, the resistivity decreases until no more oil is produced after approximately 170 min of flooding. Contrastingly, the runs with nanofluid are found to enhance oil recovery, consequently lowering the resistivity below the runs with only brine. An anomaly can be observed in the resistivity data between 195–205 min for the 1 wt% SiO_2_ run, in which the resistivity is found to rise sharply and then decrease back to the original trend. This is most likely the displacement of a large oil zone, revealing resistivity to rise as the oil pocket begins to flow and then decrease as it leaves then sensing zone. This assumption is also confirmed by the tomograms (Section 3.1) which reveal a large zone of oil being displaced at the same time.

### 3.3. Oil Recovery Rate

The slope function in excel was utilized in order to compare local recovery rate for the various runs. The charts reveal information on the rate of oil recovery since they describe changes in local resistivity, and resistivity in this case is solely a function of oil present in the sensing zones. Since the initial resistivities and oil saturations could not be fixed at exactly the same values, rather than comparing the actual values of resistivity against time, the gradient of the trends in resistivity are compared. Figure 10, Figure 11, Figure 12 and Figure 13 to show the effects of nanofluids on the local recovery rate when compared with brine-only runs. The slope was calculated between each data point (every 5 min) and presented in the form of a bar chart so that both negative and positive fluctuations could be observed, thus allowing for oil entering and leaving the sensing zone to be accounted for. Figure 10 and Figure 11 reveal the oil recovery rate for both the brine-only runs to stagnate after approximately 170 min. On the contrary the runs involving nanofluids reveal large fluctuations in gradients after 170 min highlighting the ability of SiO_2_ nanofluids to release residual oil zones and prolong the recovery process. 

### 3.4. Cumulative Oil Recovery

The oil recovery for each run is plotted in Figure 14 as a function of injected brine volume (with and without nanoparticles) related to percentage of pore volumes. The results reveal the nanofluids to be slightly more efficient than the other runs using only brine, with the 0.5 wt% SiO_2_ nanofluid resulting in an additional 2.4% (4.6 mL) oil recovery, and the 1.0 wt% run allowing for an additional 4.3% enhanced recovery (8.3 mL) (Table 4). The nanofluid pulsations were introduced after approximately 0.8 PV of brine flooding, and the nano-EOR effect was observed after approximately 1.35 PV of nanofluid pulsations in both cases. In contrast, the cumulative oil recovery during the brine-only runs were seen to plateau after around 1.2 PV and became stagnant thereafter. Both the resistivity plots and tomograms confirm the results observed in the cumulative oil recovery plot by revealing rapid changes at the same period (180–240 min) and are related to enhanced oil recovery. Therefore, ERT was found to be successful in relating changes in local resistivity to the overall oil recovery. The underlying reasons for SiO_2_ nano-EOR effects are widely studied in literature and are believed to be a combination of viscosity enhancement of the displacing phase, IFT reduction, and temporary log-jamming.

An uncertainty analysis was also conducted to account for the degree of uncertainty during the experimental runs and to highlight the repeatability of the results for this novel setup. The vertical error bars for cumulative oil recovery represent the standard deviation of the data set across three experimental runs for each scenario and reveal the sand-pack system to yield very repeatable results with errors of less than 2%. All the runs with no nanoparticles showed recovery to stagnate after around 1.2 PV, which is in line with resistivity plots revealing the same stagnation. The horizontal error bars account for the small change in pressure drop over time as the fluid leaves the drainage bags. The fluid level was found to decrease by 7 cm which corresponded to a negligible change of 0.9 psi or an error of 4% over the duration of the experiments. An error was also attributed to the oil residue left behind in the tubing at the outlet of the sand-pack which was quantified by weighing the tubing section before and after each run. The error was found to be negligible with an average of less than 0.5 g of oil residue left behind, corresponding to a negligible error of 0.06%.

### 3.5. Recovery Mechanisms

The effective viscosity was measured using an Anton Paar MCR 301 rheometer at 25 °C, and the interfacial tensions were measured using a KSV CAM 200 optical tensiometer. An IFT analysis was done in order to determine its influence as a recovery mechanism. The IFT was measured for the brine-mineral oil system at room temperature with a base value of 41 mN/m. The presence of SiO_2_ nanoparticles was found to reduce the interfacial tension between the oil and aqueous phases with an increase in concentration resulting in a reduction in interfacial tension, a trend similar to those found in literature. The sand-pack flooding results also reveal the recovery efficiency to increase slightly with nanoparticle concentration as seen in Figure 14 and IFT reduction seems to be a good candidate as the dominant recovery mechanism (Figure 15), although log-jamming may account for some additional recovery. However, quantifying the extent of recovery from log-jamming is difficult as the nanoparticles or pore throats are not visible by ERT, but the results would suggest log-jamming played a part as the IFT was approximately the same when going from 0.5 wt.% to 1.0 wt.%, however there was small trend in oil recovery observed showing 1.0 wt.% SiO_2_ to be slightly more effective.

The dynamic viscosities of brine and different SiO_2_ nanoparticle concentrations were considered in Figure 16. As predicted, the concentration of nanoparticles had an ignorable effect on viscosity. A slight increase in viscosity can be observed after the addition of nanoparticles to the brine solution from 0.818 mPa·s to 1.08 mPa·s after the addition of 1 wt.% SiO_2_. However, such a small change cannot be attributed to any substantial EOR effects. 

## 4. Conclusions

A novel low-budget sand-pack setup and effluent collection system was developed during the scope of this research and showed to yield similar results to those in literature. Tomography (ERT) showed to be an effective tool in monitoring multiphase flow behavior during flooding experiments to investigate oil recovery. The tomograms also crucially reveal the effect of SiO_2_ nanofluids on residual oil zones and the dynamics of flow behavior. The oil recovery results reveal the presence of SiO_2_ nanoparticles to enhance recovery during all runs. The difference in recovery from the 0.5 wt% and 1.0 wt% concentrations was not substantial but there was a slight trend observed showing an increase in concentration yielded approximately an extra 2% recovery. This suggests that log-jamming played a role in the recovery process as the concentration increased, because the IFT was almost the same when comparing the 0.5 wt.% and 1.0 wt.% concentrations. However, quantifying the extent of recovery from log-jamming is not possible due to limitations in ERT’s spatial resolution. The plots of local resistivity reveal large differences between nanofluid and brine-only runs with the resistivity trends stagnating during the brine-only runs after approximately 170 min. The slope function technique was introduced as a method of comparing the effects of nanoparticles on local oil recovery rate since the observed changes in resistivity during nanofluid runs were related to oil zones being released. Conductivity profiles along the central x-axis against time to show the effects of nanoparticles and highlight the physical understanding of the process. The changes in oil recovery by SiO_2_ nanofluids are highlighted by the arrows to show the increases in conductivity along the central x-axis of the tomograms as oil displacement efficiency increases, which weren’t present during the brine only runs. Furthermore, the flow behaviour was slightly altered with more pronounced peaks in conductivity after the introduction of nanofluids. A recovery analysis was also performed to determine the underlying reasons behind the observed EOR-effects. Further research involves focusing on a comparative study using different nanoparticles and introducing surfactant and polymer blends. Additionally, the ability of ERT as a monitoring tool is being considered for use in a more realistic core-flooding scenario.

## Figures and Tables

**Figure 1 sensors-19-03036-f001:**
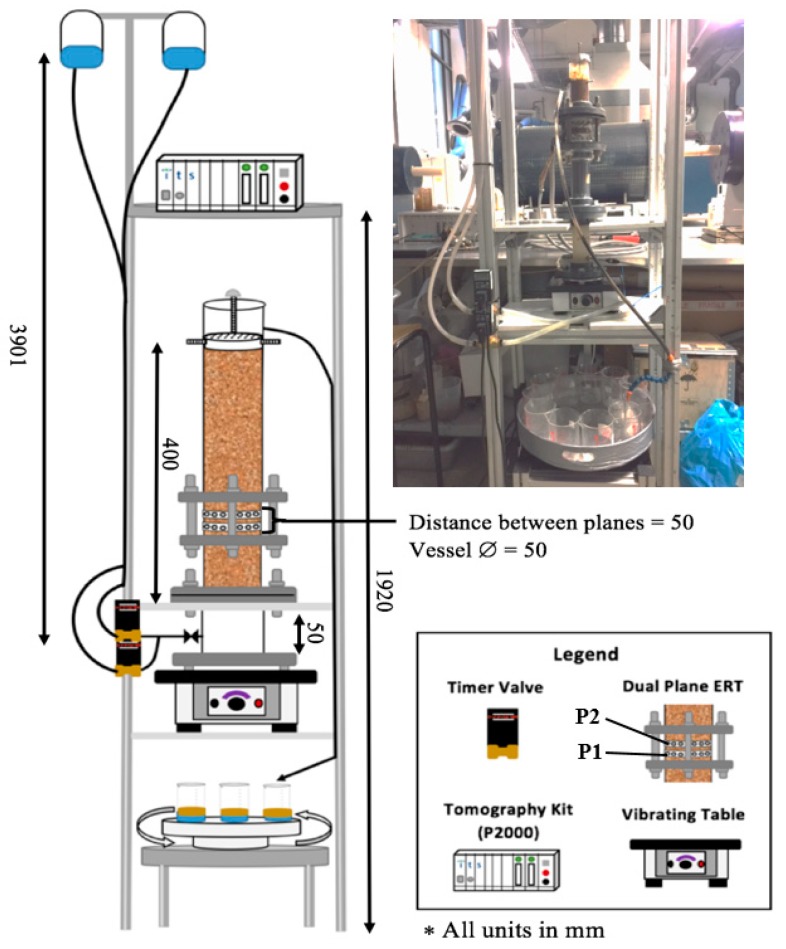
Sand-pack setup for nano-EOR experiments, including a gravity feed and effluent collection system. Where P1 and P2 represent the lower and upper sensor planes, respectively.

**Figure 2 sensors-19-03036-f002:**
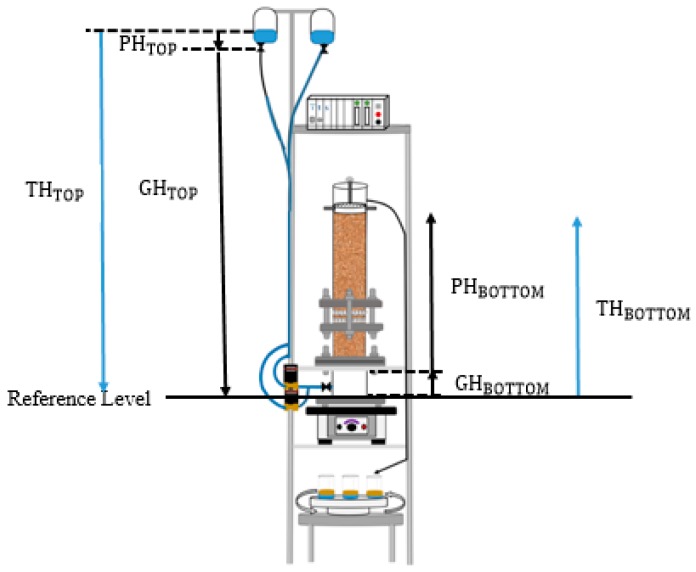
A model to explain fluid flow from gravity drainage system through the sand-pack taking into account the total heads at the top and bottom of the system. Where TH represents the total head and is the summation of GH, the gravitational head and PH, the pressure head.

**Figure 3 sensors-19-03036-f003:**
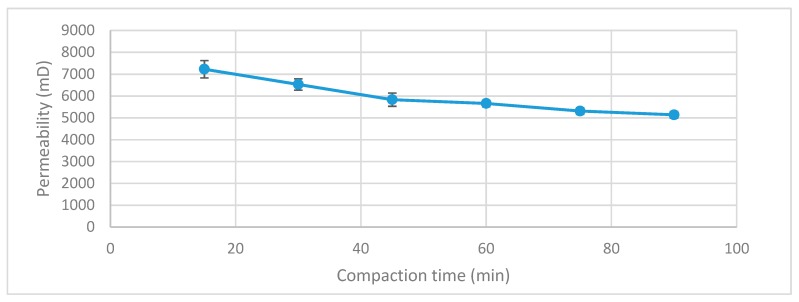
The compaction time against permeability tested in 15 min intervals for 3 runs at each compaction time, revealing the high repeatability of the packing method especially after 1 h of packing.

**Figure 4 sensors-19-03036-f004:**
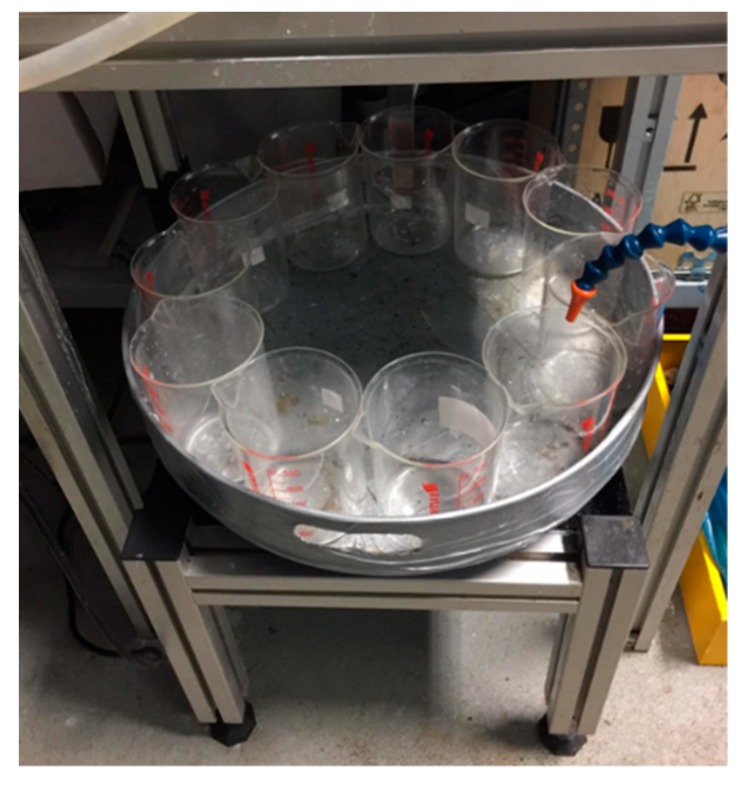
Effluent collection table with beakers arranged on the top capable of rotating automatically in timed increments.

**Figure 5 sensors-19-03036-f005:**
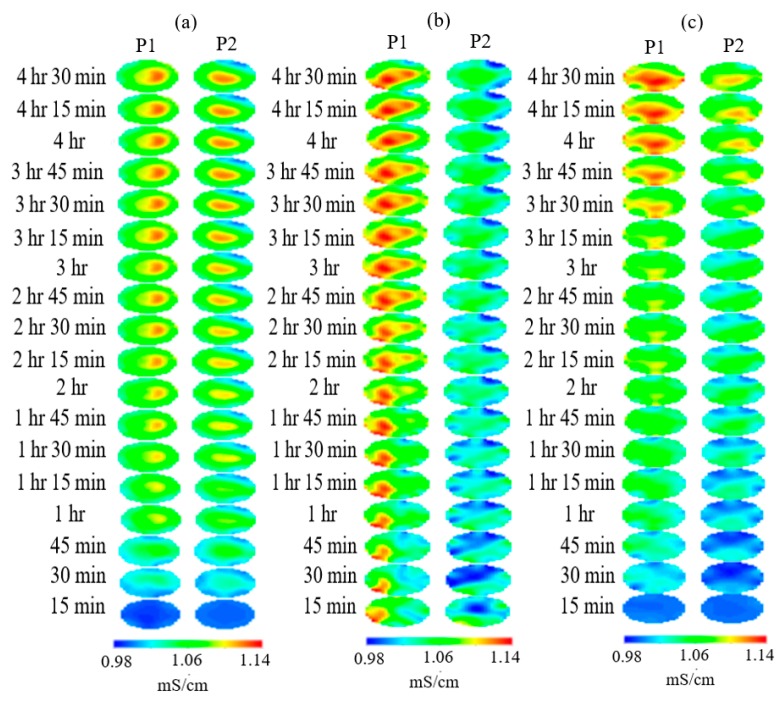
The dual plane tomograms for the flooding runs no. 1 involving (**a**) no NF (**b**) 0.5 wt% SiO_2_ and (**c**) 1 wt% SiO_2_ revealing the effect of 10 nanofluid pulsations after 1 h and 15 min of water-flooding.

**Figure 6 sensors-19-03036-f006:**
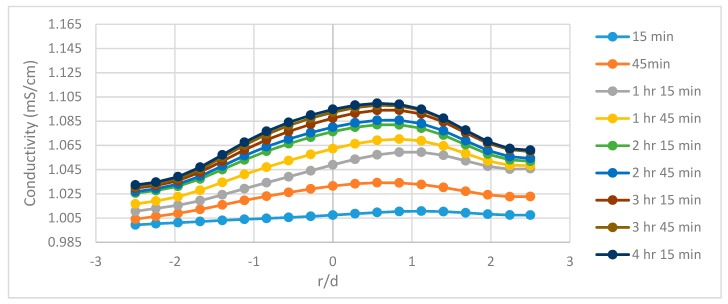
The conductivity profiles for P1 during run no. 1 with no nanoparticles, taking the central horizontal region of the tomograms into account.

**Figure 7 sensors-19-03036-f007:**
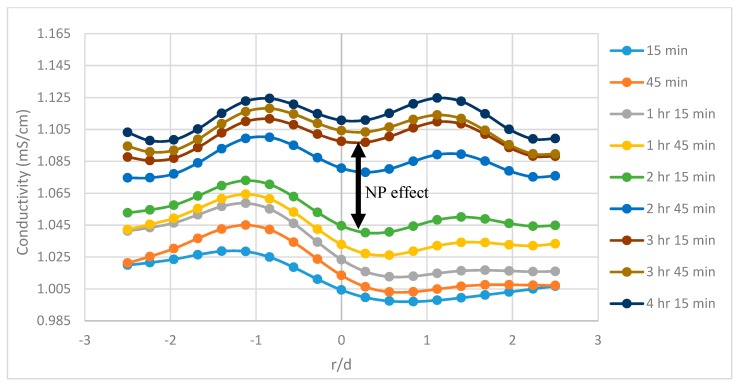
The conductivity profiles for P1 during run no. 1 with 0.5 wt.% SiO_2_, taking the central horizontal region of the tomograms into account. The effects of nanofluid are highlighted with an arrow.

**Figure 8 sensors-19-03036-f008:**
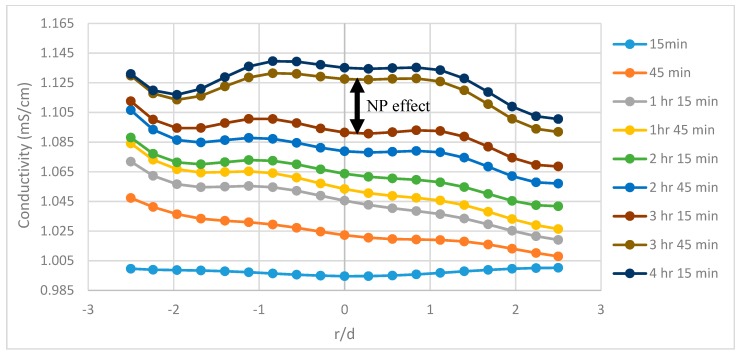
The conductivity profiles for P1 during run no. 1 with 1.0 wt.% SiO_2_, taking the central horizontal region of the tomograms into account. The effects of nanofluid are highlighted with an arrow.

**Figure 9 sensors-19-03036-f009:**
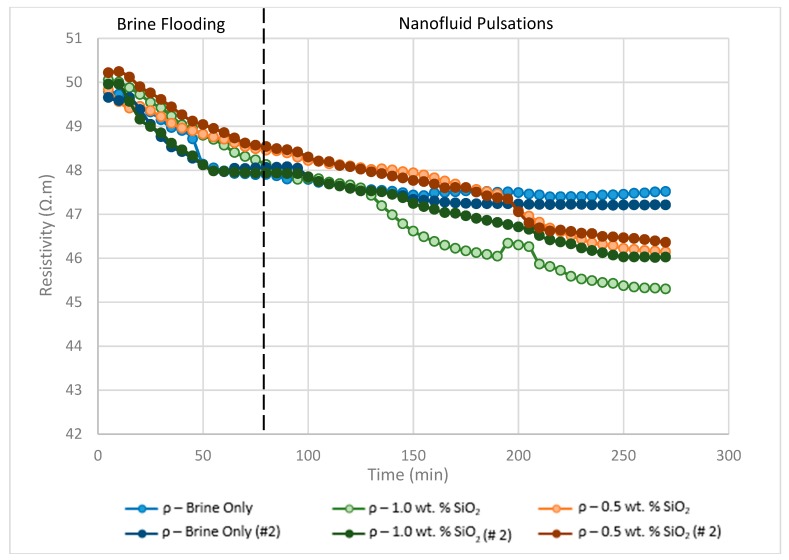
Plot of average resistivity vs oil recovered as function of time for runs no. 1 and 2 for brine flooding and both 0.5 wt% and 1.0 wt% SiO_2_ nanofluid concentrations (see Table 4).

**Figure 10 sensors-19-03036-f010:**
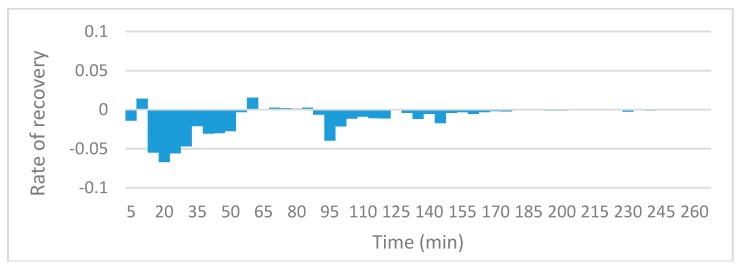
The slope function method to reveal the trend in local resistivity during brine-only run no.1.

**Figure 11 sensors-19-03036-f011:**
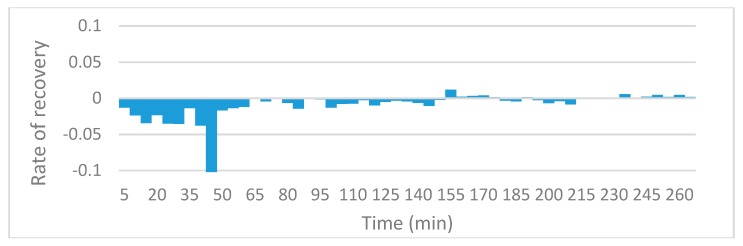
The slope function method to reveal the trend in local resistivity during brine-only run no. 2.

**Figure 12 sensors-19-03036-f012:**
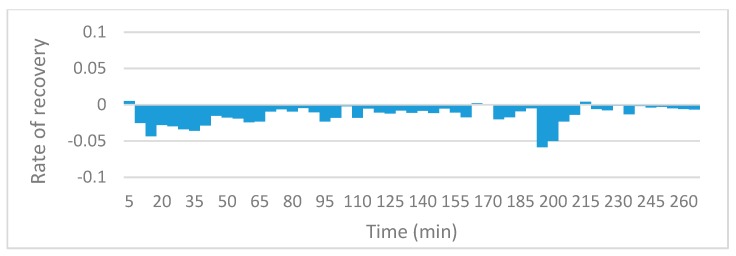
The slope function method to reveal the trend in local resistivity during the 0.5 wt% SiO_2_ run no.1.

**Figure 13 sensors-19-03036-f013:**
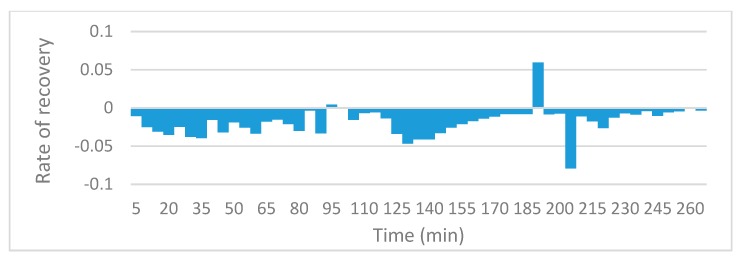
The slope function method to reveal the trend in local resistivity during the 1.0 wt% SiO_2_ run no.1.

**Figure 14 sensors-19-03036-f014:**
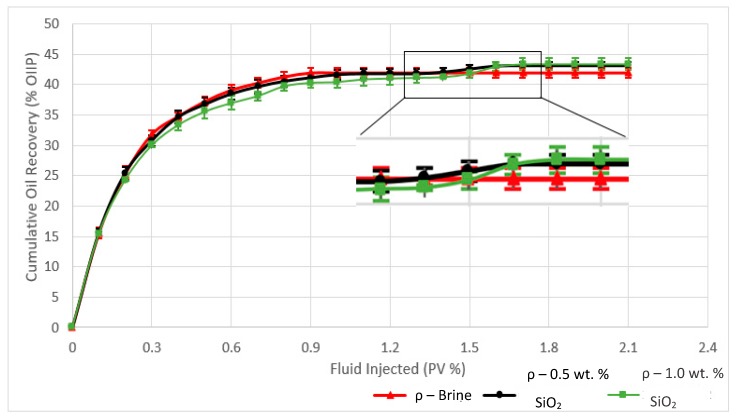
Cumulative oil recovery as a function of injected brine volume (with and without nanoparticles) related to pore volume percentage for the 3 flooding scenarios. The highlighted section reveals the duration of the flood where the nano-EOR effects can be observed.

**Figure 15 sensors-19-03036-f015:**
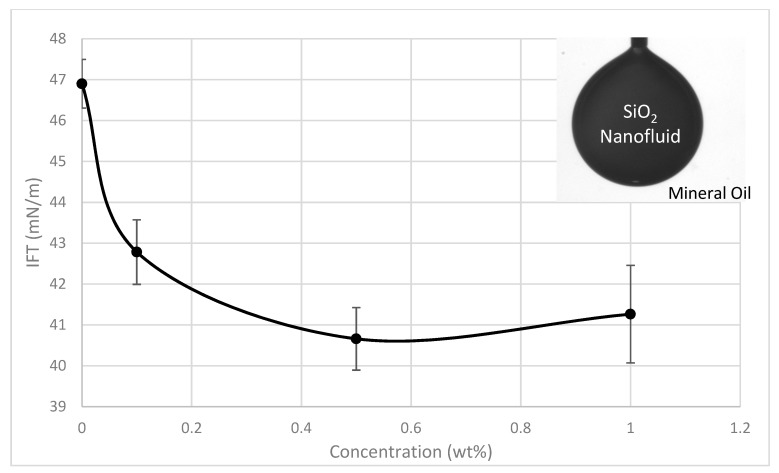
Interfacial tension of brine and various SiO_2_ nanofluid concentrations.

**Figure 16 sensors-19-03036-f016:**
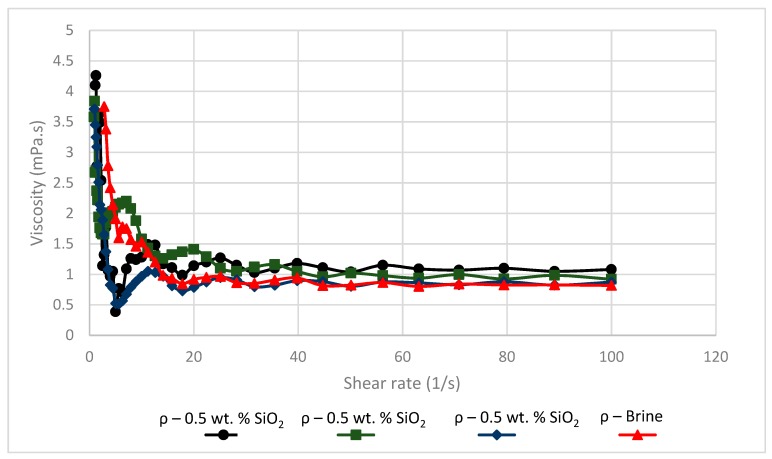
Viscosity against shear rate for brine and various SiO_2_ nanofluid concentrations.

**Table 1 sensors-19-03036-t001:** The parameters for the effluent collection table.

Effluent Collection Table Parameters							
Product Model	Diameter (mm)	Height (mm)	Transmission Gear Material	Control Method	Speed Range (Sec/Rev)	Net Weight (Kg)	Handling Load (Kg)
MT370L20	370.8	8	POM + Metal	Infrared + Bluetooth	15–31.5	5.5	20

**Table 2 sensors-19-03036-t002:** The conductivities of flooding fluids both before and after the addition of tracer NaCl solution.

Fluid Type	Volume (mL)	Pre-Mixing Conductivity (mS/cm)	Mixing Brine Volume and Conductivity	Post Mixing Conductivity (mS/cm)	
				Tracer	Probe
DI Water	2000	0.033	100 mL @ 0.6 mS/cm	0.061	0.059
0.5 wt% SiO_2_	1000	0.041	100 mL @ 0.025 mS/cm	0.060	0.058
1.0 wt% SiO_2_	1000	0.044	100 mL @ 0.22 mS/cm	0.060	0.061

**Table 3 sensors-19-03036-t003:** The parameters used to calculate the pressure drop and permeability.

Sand-Pack Parameters	
Pressure in (psi)	19.9
Pressure out (psi)	14.9
Distance (cm)	40
Area (cm^2^)	667.59
Discharge (mL/min)	2
Viscosity (Pa·s)	0.89
Permeability (mD)	5225

**Table 4 sensors-19-03036-t004:** Summary of sand-pack experiments where the initial water and oil volumes are fixed at 93.2 mL and the initial oil volume is 186.4 mL, respectively. Where %PV is the percentage of pore volume, OIIP is the oil initially in place, WF1 is the water-flooding stage, and NF represents the nanofluid pulsations.

Fluid Type	Initial Oil Saturation, % PV	Run	Oil Recovery, % OIIP		Residual Oil Saturation, % PV		Total Recovery, % OIIP
			WF1	NF Pulsations	WF1	NF Pulsations	
Brine	66	1	42.6	-	37.8	-	42.6
		2	40.9	-	39.0	-	40.9
		3	42.0	-	38.3	-	42.0
		Mean	41.8	-	38.4	-	41.8
0.5 wt% SiO_2_	66	1	41.0	2.8	38.9	37.0	43.8
		2	42.0	2.0	38.3	36.9	44.0
		3	42.8	2.3	37.8	36.2	45.1
		Mean	41.9	2.4	38.3	36.7	44.3
1.0 wt% SiO_2_	66	1	40.7	4.1	39.1	36.4	44.8
		2	40.8	3.8	39.1	36.5	43.0
		3	41.2	5.2	38.9	35.4	45.4
		Mean	41.4	4.3	39.0	36.1	44.7

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
