# Peer review of "Nanoparticle Assisted EOR during Sand-Pack Flooding: Electrical Tomography to Assess Flow Dynamics and Oil Recovery"

_sensors, 2019, doi:10.3390/s19143036_

Round 1

Reviewer 1 Report

As the authors mention by themselves, the studied phenomenon is not new. They describe a new test facility that combines a sampling and dosing system to measure the enhancement of oil recovery by the addition of nano-particles with an ERT system. In the very end, they don't really take benefit from the spatial distributions they measure with this system, but rather stay on the level of an evaluation of the average normalized electrical conductivity, providing I have understood everything correctly. I think tha paper can be published after some improvements, but it is clearly no blockbuster. Here my particular comments:

To my feeling, it would be better to speak about “pulses” of injected liquids instead of “pulsations” of injected liquids. May be one more short sentence of explanation would be helpful in the vicinity of line 91, such like: “the brine with and without nanoparticles was injected by pulsing the solenoid valve. The valve was opened for periods of ?? s once in ?? s, during which portions of about 10 ml were released from the medical drainage bags.”

What is the purpose of the calculation given in eq. (1) to (4)? It seems quite trivial.

Line 119: “In total 119 the gravity feed setup was £140” --> unnecessary information.

Line 161: “purchased from a Chinese manufacturer (model JT-51B)” --> mention name of manufacturer of delete.

Typo on Figure 3: “Comapction time” change to “Compaction time”.

Line 224: “Next the nanofluid pulsations were introduced” --> shouldn’t it be “particles” instead of “pulsations”?

Figure 6. I don’t understand why the authors call the plot “Plot of resistivity vs oil recovered...”, when it is in fact “Plot of resistivity vs time...”?

Section “3.3. Slope Function Method” and related figs. 7 – 10 do not add much to the already discussed results presented in Fig. 6 and can (should) be omitted.

Line 342: “The oil recovery for each run is plotted as a function of pore volumes injected in fig 11.” I don’t fully understand. Shouldn’t it be formulated like this: “The oil recovery for each run is plotted in fig 11 as a function of the injected brine volume (with and without nanoparticles) related to pore volumes”. Furthermore, is the x-axis showing the ratio of injected volume to pore volume in absolute terms or in percent? Please review the figure subscript as well.

The oil recovery seems to have a quite strong spatial non-uniformity with a maximum in the center of the column, which is not discussed by the authors. They seem to use only an average value of the conductivity for their conclusions. It remains unclear what the tomographic ability of the applied ERT adds to the physical understanding of the process.

Author Response

Thanks for your helpful input and feedback. Attached please find the point-by-point responses. I’ve addressed a couple of points below which may need a bit more clarification:

All of the spelling mistakes have been corrected, and information was added to clarify points which weren’t explained properly. I do agree more benefit was taken from the average normalised conductivity and to a lesser extent the spatial distributions. So I’ve added conductivity profiles along the central x-axis against time to show the effects of nanoparticles and highlight the physical understanding of the process. This gives a better understanding of the flow dynamics and how the nanoparticles changed the flow behaviour and oil recovery. The changes in oil recovery are highlighted by the arrows to show the increases in conductivity along the central x-axis of the tomograms as oil displacement efficiency increases, which weren’t present during the brine only runs. Additionally, the flow behaviour was slightly altered with more pronounced peaks after the introduction of nanofluids.

Kind regards,

Phillip Nwufoh

Reviewer 2 Report

The article raises many serious doubts in terms of both relevance and science.

The subject of the article suggests that the objective of research will be the development of a new tomographic method or algorithm. It turns out that a standard tomograph was used (line 98) and the aim of the research was to: „… construct a new sand-pack setup which incorporates ERT sensors around the sand-pack.” (line 77)

In the subject suggesting research in the field of tomography, the basic disadvantage is the use of a tomograph that has not been developed or improved by researchers - but just purchased (line 98). The authors do not bring anything new in the field of tomographic imaging methods in terms of the construction of new equipment. They do not propose new algorithmic or computational methods.

The authors have also not developed a new chemical method for recovering crude oil. It seems that the novelty was how to use an electrical tomograph to imaging the inside of a sand-pack cylinder. However, it cannot be considered as a scientific contribution. This is rather an engineering achievement.

In line 396: “A novel sand-pack setup and effluent collection system has been developed during the scope of this research and has been shown to yield similar results to those in literature.”

Why develop another sand-pack setup, if similar results were obtained as with other methods?

Lines 136-144: What is the purpose of these calculations? Is this impossible to check empirically? Is the result not too obvious?

Line 120: What gives an information about the costs of purchasing a laboratory device? Does this translate into the cost of the method in industrial applications?

Author Response

Thanks for your helpful input and feedback. Attached please find the point-by-point responses. I’ve addressed a couple of points below which may need a bit more clarification:

I made the aims more clear by making small changes to the sentence structure, the layout of the sentence is not very clear and would indeed potentially confuse readers. I understand and agree the contribution isn’t ground breaking in terms of new methods for both oil recovery and tomographic imaging. The aims of the paper aren’t to develop a new tomographic method or to come up with a new chemical method but rather to explore the ability of tomography to show changes to local oil recovery, flow structure and flow dynamics brought about by the presence of nanoparticles, and to link changes in local recovery efficiency to overall oil recovery. No imaging technique has been used in this kind of setting before, and therefore there were many engineering problems that had to be overcome such as account for the references at each stage of flooding (which was overcome by the packing method – wet packing the sand with oil and water, allowing for the traditional drainage process to be skipped), how to ensure the resistivity contrast is a function of oil recovery (achieved by tracer dilution method to dope the fluids correctly, in addition to low budget solutions for introducing and collecting the fluids.

The changes to flow dynamics and local recovery efficiency plots when comparing brine only runs and nanoparticle are quite substantial and reveal big differences which haven’t been shown to date using ert or any other imaging technique. The ability of SiO2 nanofluids to open new flow channels and alter the flow structure was also reported now that I’ve added the conductivity profiles against time. Additionally small differences were highlighted between the 1.0wt%, and 0.5wt% concentrations when using solely nanoparticles without any surfactant or polymer blends. This suggests log-jamming would play a role at higher concentrations, however as I stated in the paper it isn’t able to be quantified or explored with only ert due to spatial resolutions. However I don’t agree the is no scientific contribution at all because the changes in flow structure, oil recovery, and resistivity/ conductivity profiles highlighted by ert reveal the nature of nanofluid flow and recovery efficiency compared with brine-only, especially when considering no other imaging techniques have been employed in the past to show such detailed temporal changes over the full duration of flooding experiments (usually lasting a few hours).

The other suggestions I agree with and have made corrections.

Kind regards,

Phillip Nwufoh

Round 2

Reviewer 2 Report

The authors' explanations are convincing and sufficient. Correction of the text was carried out in a proper, effective and thoughtful manner. In its current form, the article is suitable for publication.